# A One Health Perspective on Canine Coronavirus: A Wolf in Sheep’s Clothing?

**DOI:** 10.3390/microorganisms11040921

**Published:** 2023-04-02

**Authors:** Alessio Buonavoglia, Francesco Pellegrini, Nicola Decaro, Michela Galgano, Annamaria Pratelli

**Affiliations:** 1Dental School, Department of Biomedical and Neuromotor Sciences, Via Zamboni 33, 40126 Bologna, Italy; alessio.buonavoglia85@gmail.com; 2Department of Veterinary Medicine, University Aldo Moro of Bari, Sp Casamassima Km 3, Valenzano, 70010 Bari, Italy; francesco.pellegrini@uniba.it (F.P.); michela.galgano@uniba.it (M.G.); annamaria.pratelli@uniba.it (A.P.)

**Keywords:** coronavirus, dog, human, recombination

## Abstract

Canine coronavirus (CCoV) is a positive-strand RNA virus generally responsible for mild-to-severe gastroenteritis in dogs. In recent years, new CCoVs with acquired pathogenic characteristics have emerged, turning the spotlight on the evolutionary potential of CCoVs. To date, two genotypes are known, CCoV type I and CCoV type II, sharing up to 96% nucleotide identity in the genome but highly divergent in the spike gene. In 2009, the detection of a novel CCoV type II, which likely originated from a double recombination event with transmissible gastroenteritis virus (TGEV), led to the proposal of a new classification: CCoV type IIa, including classical CCoVs and CCoV type IIb, including TGEV-like CCoV. Recently, a virus strictly correlated to CCoV was isolated from children with pneumonia in Malaysia. The HuPn-2018 strain, classified as a novel canine–feline-like recombinant virus, is supposed to have jumped from dogs into people. A novel CoV of canine origin, HuCCoV_Z19Haiti, closely related to the Malaysian strain was also detected in a man with fever after travel to Haiti, suggesting that infection with Malaysian-like strains may occur. These data and the emergence of highly pathogenic CoVs in humans underscore the significant threat that CoV spillovers pose to humans and how we should mitigate this hazard.

## 1. Introduction

Canine coronavirus (CCoV) is an enveloped virus within the *Coronaviridae* family with a single-strand positive-sense RNA genome, approximately 27–32kb in size. Based on phylogenetic analyses and genomic structures, CoVs are currently classified within family four genera, *Alphacoronavirus*, *Betacoronavirus*, *Gammacoronavirus*, which is the largest known RNA genome, and *Deltacoronavirus*, which recognises bats, birds, and likely rodents as natural reservoirs [1]. CCoV belongs to the genus *Alphacoronavirus*, which also includes the prototype viruses: feline coronaviruses (FCoVs), transmissible gastroenteritis virus (TGEV), mink coronavirus (MinkCoV), ferret coronavirus (FRCoV), and alpaca respiratory coronavirus [2]. Two thirds of the CCoV genome from the 5′ end consists of two large overlapping open reading frames (ORFs), ORF1a and ORF1b, encoding the replicase protein, the viral RNA-dependent RNA polymerase, and viral proteases. The remaining genome from the 3′ end contains ORFs encoding the four major structural proteins, spike (S), membrane (M), envelope (E), and nucleocapsid (N) proteins, and for the non-structural proteins ORF3a, ORF3b, ORF3c, ORF7a, and ORF7b [3]. The spike protein, which assembles into trimers on the virion surface to form the distinctive “crown” appearance, is the main inducer of neutralizing antibodies and plays an important role during viral entry, mediating the viral attachment to specific host cell receptors and the fusion between the envelope and the plasma membrane [4]. The S protein ectodomain shares the same organization in all CoVs and is organized in two distinct domains: S1, the N-terminal domain (NTD) responsible for receptor binding, and S2, the C-terminal domain (CTD) responsible for fusion. A notable difference between the S proteins of different CoVs is whether they are cleaved during viral replication. Most *Alphacoronaviruses* and *Betacoronaviruses*, with some exceptions, share an uncleaved protein, but the S protein of *Gammacoronaviruses* and some *Betacoronaviruses* is cleaved into S1 and S2 domains by a Golgi-resident protease [5]. The most represented and abundant viral structural protein is the M protein, a type III glycoprotein consisting of a short amino-terminal ectodomain, a triple-spanning transmembrane domain, and a long carboxyl-terminal inner domain [6]. It contributes to core stability and can play a role in neutralizing viral infectivity in the presence of complement [7]. The small E protein was recognised as a structural component of the CoVs and is thought to be important for viral envelope assembly [8]. The N protein is a basic phosphoprotein that forms the helical nucleocapsid that binds to the viral RNA and modulates RNA synthesis [4]. The additional ORFs encoding non-structural proteins vary among different CoVs in number, nucleotide sequence, and gene order [9]. Their role is sometimes unknown and most of them are not essential for viral replication but may play a role in virulence [10].

## 2. Canine Coronavirus: Light and Shadow

CCoV is considered one of the main pathogens responsible for enteritis in dogs, wolves, foxes, and other canine species [11], but it can also affect other animals thanks to its ability to undergo recombination that ensures the proliferation of new strains with selective advantages over the parental genomes and its ability to easily cross interspecies barriers. This host species extension, as well as the variation in the cell tropism and the pathogenicity of the virus, is mainly related to the variability of the S protein, which is responsible for the emergence of new virus strains, serotypes, and subtypes [12].

Starting from its first description in 1971 during an epizootic infection in a canine military unit in Germany [13], CCoV rapidly spread worldwide and today appears to be enzootic with a variable prevalence ranging from 6.25% to 72.5% [14,15]. Although in the past CCoV has been overlooked and vaccination is still not recommended due to the absence of an effective challenge model, this virus is involved in the onset of moderate-to-severe enteritis. Dogs of all breeds and ages are susceptible to infection and, although the virus alone can be responsible for epizootics mainly in kennels and animal shelters, fatal infections are unusual unless mixed infections and/or overcrowding/unsanitary conditions exist [3]. Nevertheless, in the last few decades, CCoV has undergone mutations/recombination over time, changing tissue tropism and virulence and generating new genetically divergent strains, some of which possess pathogenic potential [16]. After highlighting point mutations affecting the M gene and multiple regions of the viral genome including ORF1a, ORF1b, and ORF5 in several CCoVs detected in pups with diarrhoea in Italy, the phylogenetic analysis on the inferred aa sequence of a region encompassing about 80% of the S gene of one of these strains (strain Elmo/02) clearly showed that strain Elmo/02 segregates with FCoVs of type I (sharing about 81% identity) rather than reference CCoVs and FCoVs of type II (sharing about 54% identity) [17,18,19]. Other features differentiate Elmo/02 from classical strains: (i) a potential cleavage site in the S protein shared with *Betacoronavirus* and *Deltacoronavirus*; (ii) an additional ORF (referred as ORF3) 624 nt in length and encoding a putative 207aa-protein—likely secreted from the infected cells as no transmembrane region was found—that is completely absent in FCoV type I and of which only remnants remain in the genomes of CCoV type II and TGEV [20]. Based on these observations, two genotypes of CCoV designated as CCoV types I and II were proposed as observed from the genomic analysis of the genes encoding the main structural proteins of the virus [18,21,22]. Elmo/02 was designated as the prototype of the newly recognized CCoV type I, and the reference strains (i.e., Insavc-1, K378) as CCoV type II [18]. Most detected CCoVs are type II viruses that are also easily cultivated in canine and feline cell lines. On the contrary, CCoV type I, to date, has been only detected by reverse transcription-polymerase chain reaction (RT-PCR) and quantitative PCR (q-PCR) [16].

Reports on the emergence of new CCoV strains have always occurred and have been constant over time. Wesley [23] demonstrated that the N-terminus of the S gene of strain UCD-1 is more closely related to TGEV than to CCoV, and Naylor et al. [24] identified a divergent CCoV strain, UWSMN-1, in Australia, characterized by the presence of gradually accumulating point mutations randomly interspersed throughout its genome. In 2002, a new CCoV, BGF10, with a highly divergent region at the amino-terminal domain of the M protein and a long non-structural protein 3b—250 aa long and associated with virulence in other CoVs—was isolated in the United Kingdom [25]. Possible recombination events also occurred in certain CCoV strains detected in a fatal outbreak of gastroenteritis in Swedish dogs. These virulent strains displayed the 5′ end and 3′ end of an S gene closely related to CCoV type I and CCoV type II, respectively, suggesting a possible recombination event between CCoV genotypes [26].

The most striking example of a hypervirulent strain was observed in 2005 in Italy and later in other countries. The detected strain, CB/05, designated pantropic CCoV type II, was associated with the onset of severe and haemorrhagic gastroenteritis, which was sometimes fatal [27,28,29,30]. Interestingly, the CB/05 strain—detected in other organs including the brain in addition to the intestines—has a degree of aa identity to CCoV type II in the 3′ end of the genome, although the S gene displayed the highest identity to FCoV type II, strain 79-1683. In addition, a 38-nt deletion in ORF3b was identified as a genetic marker of this pantropic CCoV type II strain, but no obvious genetic signatures responsible for the switch in pathogenicity were found [31].

A genetic analysis of the accessory gene ORF3 highlighted that TGEV originated from CCoV through cross-species transmission, pointing out that the evolution of TGEV, FCoVs, and CCoVs are closely related to each other. In fact, not only did CCoV give rise to FCoV type II through double homologous recombination events between the S and M genes [32], and recombination events with an ancestral CoV might have given rise to the appearance of FCoV type I and CCoV type I [19], but a further recombination between CCoV type II and TGEV has occurred in the very 5′ end of the S gene, generating a back recombination event with the appearance of a TGEV-like CCoV (Figure 1). Phylogenetic analysis showed that TGEV-like CCoVs formed a monophyletic group with the same cluster of TGEV and porcine respiratory CoV in the N terminus of the S gene; however, in the C terminus, they clustered together with CCoV-II isolates and separately from porcine respiratory and enteric CoVs. The recombinant TGEV-like virus was detected in the internal organs and in the intestinal contents of infected pups which died with acute gastroenteritis, underlining the pathogenic role of this virus [33]. In addition, during 2001–2008, the distribution of the TGEV-like CCoVs was assessed in the canine populations of different geographic areas of Europe, and approximately 20% of positive CCoV samples were characterized as TGEV-like CCoV, with a detection rate which varied according to geographic origin. The confirmation of TGEV-like CCoV circulation among dog populations has epidemiological implications for prophylaxis programs in dogs based on the administration of CCoV vaccines prepared with the classical CCoV type II strains, which may not provide protection against TGEV-like CCoV infection and should draw attention to the pathogenetic and epidemiological role of these recombinant CCoV viruses [34]. The identification of a TGEV-like CCoV characterized by a potential double recombination through partial S-gene exchange with TGEV induced the proposal of a new classification for CCoV type II, which is now further divided into two subtypes: CCoV type IIa (including classical enteric CCoVs) and CCoV type IIb (including TGEV-like CCoVs) [33].

Recently, a study on the characterization of an old CCoV strain (A76 CCoV) was reported [35]. The virus was isolated in 1976 from a closed breeding colony of beagles at the James A. Baker Institute for Animal Health (Cornell University, Ithaca, NY) from dogs with enteritis, although new-born pups exhibited additional clinical signs associated with significant morbidity and abortion was sometimes observed [36]. The virus possesses distinct characteristics as compared to CCoV type II. Efficient replication of the A76 strain is restricted to canine cells because this strain only uses the canine aminopeptidase N receptor in contrast to classical CCoV type II viruses that can use both the canine and feline aminopeptidase N receptor. Moreover, genetic analysis highlighted that the S gene is a recombinant between CCoV type I and CCoV type II in the NTD and CTD of the S1 subunit, and this feature is probably responsible for the different host cell tropism observed [35]. The strain A76 clustered with the B906_ZJ_2019 strain recently detected in China (93.24% nt similarity), forming a single clade in the CTD. This suggests the presence of an additional recombination site in strain B906_ZJ_2019, although this has not yet been confirmed, which, nevertheless, points out that recombination events occur more frequently than expected [14]. In line with the characterization of the A76 and Chinese B906_ZJ_2019 strains, the classification described above into two genotypes should be revised. Phylogenetic analysis revealed a third subtype, the CCoV type II variant, which includes the A76 and B906_ZJ_2019 strains that differ from CCoV type IIa and CCoV type IIb and might be evolved from recombination events between CCoV type I and CCoV type II [14].

Recently, Sha et al. [37] analysed several CCoV strains which showed unique variants of the S1 subunit NTD. Among these, the CCoV type IIb strain SWU-SSX7 clustered with the Chinese ferret badger strain and showed a complex evolutionary process being characterized by recombination events in the S1 subunit between CCoV type IIa and CCoV type IIb, with a breakpoint starting at 2141 nt. This is a further report on the emergence of new CCoV recombinants that can lead to changes in the genetic characteristics and biological properties of CCoVs, and to a gradual expansion of the host range of these viruses.

## 3. Pathobiology of CCoV Infection in Dogs

### 3.1. Clinical Signs

Clinical disease caused by CCoV in dogs is extremely varied, still not fully understood, and with a clinical evolution influenced by many factors, especially by the genetic characteristics of the strains. The virus is responsible for mild-to-moderate enteritis but young pups may develop severe clinical signs, especially when infections with other pathogens occur simultaneously. In addition, coinfections with intestinal viruses that increase the pathogenicity of CCoV, such as canine adenovirus type 1 (CAdV-1) or canine distemper virus (CDV), have been reported [38,39]. Dual infections with canine parvovirus type 2 (CPV2) are often detected and are especially severe, but CCoV can also enhance the severity of a subsequent CPV2 infection [3]. The main source of infection is represented by faeces, and infection is characterized by oronasal transmission. Clinical signs appear after a short incubation period and are characterized by vomiting, which usually subsides after the first day of illness, and diarrhoea. The elective site of virus replication is the villus tips of the enterocytes of the small intestine, where CCoV is responsible for lytic infection followed by desquamation and shortening of the villi. Virus replication results in malabsorption and the deficiency of digestive enzymes, and consequently in diarrhoea which may already arise 18–72 h post infection and generally persist for 6–9 days when the production of local IgA antibodies restricts both CCoV diffusion within the intestine and the clinical evolution of the disease. However, sometimes, infected dogs may shed the virus intermittently for as long as 6 months after the disappearance of clinical signs [16]. The faeces may be mucous or watery but it is rarely haemorrhagic, except if poor hygienic conditions and coinfection exist; in puppies, the clinical signs can be characterized by dehydration, sensory depression, and anorexia, even in the absence of hyperthermia and leukopenia [3]. Mixed infections with both genotypes (CCoV type I and CCoV type II) are commonly reported, but failure to isolate CCoV type I in cell cultures hampers the acquisition of key information on the pathogenetic role of CCoV infections in dogs and makes it difficult to evaluate the epidemiological and immunological characteristics of CCoV infection [40]. Although serological investigations on the spread of CCoV have widely demonstrated that the virus is present throughout the world, few reports of gastroenteritis in dogs clearly attributable to CCoV are reported and few strains have been cultivated in cell culture [28]. The low detection rate is mostly attributable to the low viral titre excreted in the faeces, especially in long-shedding infected pups, to the low stability of the enveloped virus in the environment, and to the low stability of the RNA [41].

### 3.2. Emerging CCoV Strains and Associated Diseases

In addition to the classic (mild) enteric CCoVs, several outbreaks of infection characterized by severe clinical signs followed by fatal outcomes are reported. The pantropic CB/05 CCoV type IIa strain isolated in 2005 in Italy caused a systemic disease characterized by high fever, anorexia, lethargy, haemorrhagic diarrhoea, vomiting, leukopenia, and neurological signs with ataxia and seizures. This virulent strain was unexpectedly detected at high titres in the lungs, spleen, liver, kidney, and brain, in addition to the intestinal content [28]. Moreover, the detection of TGEV-like CCoV in 2008 in Italy corroborated the observations on the ability of CCoVs to cause more serious infections than the classic enteric forms. TGEV-like CCoV strains were detected in the faecal samples, in the intestinal contents, and in the internal organs of pups which died with gastrointestinal signs [33]. Despite the fact that infected pups were coinfected with CPV2 that could have played a relevant role in TGEV-like CCoV spreading to the internal organs, this must be a wake-up call on CCoV’s pathogenetic evolutionary ability. The detection of virulent CCoV strains associated with the ability to spread from the enteric tract to other internal organs strongly corroborates the ability of CoVs to modify their virulence and organ tropisms, suggesting the importance of an investigation of the molecular basis of these mechanisms through the assessment of a reverse-genetics system similar to that established for feline infectious peritonitis virus [28,42]. The meaning of all these data is not yet fully understood, but they certainly raise several questions on the pathobiology of CCoVs, both in terms of virus evolution and immunization plans. The genetic differences observed in the spike protein between classical enteric strains and recombinant CCoVs may have particular implications for the efficacy of vaccines. Common vaccines are set up with reference CCoV type IIa strains, and dogs vaccinated with these vaccines might be susceptible to infection or disease caused by TGEV-like CCoVs. Only a vaccination trial with a commercial vaccine and the subsequent challenge with recombinant TGEV-like CCoV could confirm and/or assess the level of cross-reactivity between CCoV type IIa and TGEV-like CCoV type IIb, since a single previous study showed a poor cross-reactivity [33]. Therefore, despite these recombinant viruses being detected in many countries, few data are available in the literature on the clinical features of infection caused by this virus and only some aspects have been clarified. It is nevertheless worthwhile to carefully monitor these viruses to better understand the clinical, pathogenetic, and epidemiological implications of these different genotypes. As a result of the relatively high mutation frequency of RNA viruses, CCoVs are able to rapidly adjust to negative pressures such as those presented by the immune system. Recombination events affecting CCoVs could clarify the evolutionary processes leading to the onset of new strains, as has been observed for severe acute respiratory syndrome (SARS)-CoV-1 and SARS-CoV-II and for CCoV type IIa (CB/05) and CCoV type IIb (TGEV-like CCoV). Many aspects of the pathobiology and evolutionary features of CoVs remain to be clarified, among which are the meaning of simultaneous infection by CCoV type I and CCoV type II, the pathogenetic role of these viruses, the immune response, and the efficacy of the currently used vaccines against emerging CCoV strains.

## 4. The Zoonotic Features of Canine Coronaviruses

CoVs easily cross interspecies barriers and undergo genetic evolution, giving birth to new strains with a selective advantage over the parental genome and characterized by dramatic changes in virulence and tissue tropism [43]. Since cross-species transmission frequently occurs and mixed infections of different genotypes exist—favouring the emergence of new viruses with increased pathogenicity and different host cell tropisms—aa changes and recombination events occurring in these genetically unstable viruses should be carefully monitored [33,44].

Before 2002, no in-depth information was available on human CoVs (HCoVs) and their zoonotic potential. The emergence of the first highly pathogenic HCoV in 2002, SARS-CoV-1, the subsequent epidemic Middle East respiratory syndrome CoV, (MERS-CoV), and the pandemic SARS-CoV-2 in 2012 and late 2019, respectively, opened “Pandora’s box” on CoVs, and researchers exploited the extensive knowledge in veterinary medicine on animal CoVs to better understand the characteristics of these viruses and to study their origin [45]. Most recently, zoonotic diseases have originated in wildlife species, favoured mainly by interfacing between humans and wildlife and by human activities and urbanization [46]. Among these, in recent years, episodes of suspected zoonotic transmission of CoVs from animals to humans have emerged, with the substantial difference that wild animals, notoriously considered to be the reservoirs of CoVs infections, were no longer involved in the transmission. In 2017–2018, a closely related CCoV, the HuPn-2018 strain, was isolated in the canine A72 cell line from nasopharyngeal swabs of children with pneumonia in Malaysia. The virus was characterized as a novel canine –feline-like recombinant virus with a unique deletion in the nucleoprotein, similar to the deletion found in SARS-CoV-1 and SARS-CoV-2 that occurred very soon after their introduction into humans [47] (Figure 2). As the main genome of CCoV-HuPn-2018 was the same as some CCoV strains, it was supposed that the virus jumped directly from dogs into people [47]. The group of Lednicky has also identified a CoV of canine origin, designated HuCCoV_Z19Haiti, closely related to the Malaysian CCoV-HuPn-2018 (99.4% identity) except for the second part of the genome from gene E and the presence of two characteristic deletions in gene N and ORF7b of the Malaysian virus [48]. Detailed phylogenetic analysis highlighted that the S gene clustered with CCoV-HuPn-2018, whilst the M gene was closer to CCoV B639_ZI_2019 [14] and the N gene clustered with TGEV, although the bootstrap values were too low to generate a strong inference [48]. HuCCoV_Z19Haiti is an example of a further recombination event affecting CCoV, highlighting once again the mutational feature of these viruses that periodically favours the emergence of new viruses crossing the species barrier and jumping into humans. This chimeric Haitian virus showed a low virulence, but this high potential for genetic recombination that ensures the proliferation of new strains should be monitored in view of a possible but not negligible hypothesis: such viruses are able to acquire new genes capable of transmitting a selective advantage over their parental genomes [48].

Furthermore, until 2021, reported human infections by CoVs were caused by *Alphacoronavirus* and *Betacoronavirus*, whilst *Gammacoronavirus* and *Deltacoronavirus* were known to infect primarily birds with some mammalian spillover [49]. Interestingly, in three plasma samples collected between 2014 and 2015 from Haitian children with acute undifferentiated febrile illness, a CoV strain related to porcine deltacoronavirus (PDCoV) was identified. Genomic analysis indicated that the virus may have been transmitted to the children through two independent zoonotic transmission events that likely occurred from the same animal, or different animals infected with a very similar virus, or may be the result of an animal-to-human transmission followed by human-to-human transmission. This was the first signalling of PDCoV infection in humans, and the presence of a unique aa sequence involving the Nsp15 and S proteins suggests a role for these aa changes both in the adaptation of the virus to a human host, and in the capacity for human-to-human transmission in *Deltacoronaviruses* [50].

The occurrence of human epidemics/pandemics caused by SARS-CoV-1, MERS-CoV, and SARS-CoV-2, the detection of CCoV-HuPn-2018 in children with pneumonia in Malaysia, and the first reports of PDCoV infection in Haitian children have raised questions about the zoonotic potential of these viruses and their ability to overcome species barriers.

## 5. Conclusions

Almost 60% of human pathogens and approximately 60% emerging infectious diseases are zoonoses, and zoonoses transmitted from animals to humans represent an increasingly global public health problem since, despite the fact that animal-to-human transmission has always occurred in the past, its frequency has increased in recent years [46,51]. CoVs possess a strong ability to overcome species barriers and to jump from reservoirs to humans, especially from bats which are considered the primary carrier and reservoir for CoVs and other several viruses. Considering the large diffusion and genetic diversity of bat CoVs, the large number of bats living in communities, and their ability to travel long distances, it cannot be excluded that new viruses may emerge in the future. This is a worrying hypothesis which, despite its impact, has not seen the implementation of any contrasting action to limit human–animal cohabitation (especially with wildlife animals). In alignment with the concept of “One Health”, the detection of Malaysian and Haitian CoVs in humans not only confirms the genetic plasticity of CoVs but should cause alarm and induce the scientific community to carry out continuous genomic surveillance both in wildlife and in companion animals to avoid the next possible spillovers (species jumps) from animals to humans and future severe heath emergencies [12,16,52]. In conclusion, the recent reports of human infections caused by canine coronavirus-like strains could be a meaningful warning: CCoVs, generally considered not particularly aggressive for dogs, could assume a relevant and non-negligible pathobiological role leading to the image of “a wolf in sheep’s clothing”.

## Figures and Tables

**Figure 1 microorganisms-11-00921-f001:**
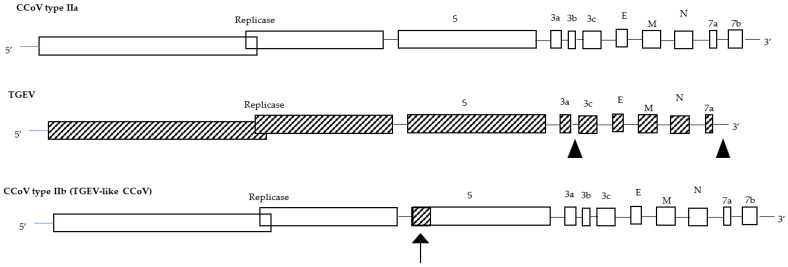
Genome organization of CCoV type IIa, TGEV, and CCoV type IIb (TGEV-like CCoV). The arrow shows the recombination site.

**Figure 2 microorganisms-11-00921-f002:**
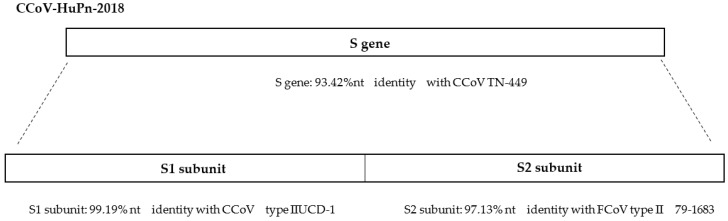
Nucleotide (nt) identity (%) of the CCoV-HuPn-2018 S gene with CCoV type II and FCoV type II [47].

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
