# Peer review of "A One Health Perspective on Canine Coronavirus: A Wolf in Sheep’s Clothing?"

_microorganisms, 2023, doi:10.3390/microorganisms11040921_

Round 1

Reviewer 1 Report

The review paper by Buonavoglia et al., present an updated overview of the role of Canine Coronaviruses as zoonotic emerging viruses. The low number of cases reported make the paper more a Commentary-like paper than a full review. The paper needs some graphics material  to explains different point of view of Authors.

Line 28: Coronaviridae Family

Line 29. One of the largest. Specifically the GammaCoV are the largest.

Line 36. 5’two-third? – One third. Use better a most accurate and scientific  description

. Canine coronavirus: light and shadow: A graphic scheme is highly recommended for explain recombination events or a phylogenetic diagram indicating CCoV type IIa and CCoV type IIb and variant third subtype

Line 174-175. Clinical disease caused by CCoV is also related to the virus strain .. please correlate to the previous paragraph

Line 194: Type 1 vs Type-I

3. Pathobiology of CCoV infection in dogs: Please split in two or three paragraphs . Too much information for just one paragraph

Line 199. The low detection rate in dogs cannot be due to the low stability of the enveloped virus in the environment. Perhaps low detection in the environment or low stability of the RNA in the labs

Please mention on this paragraph other coinfections whit enteric viruses such Canine Adenovirus, canine Circovirus, etc etc.

Line 243: delete: Despite the labile and unstable nature of RNA viruses.

Line 249. Reference is missing

Line 256 -257. Reference is need to support this commentary

Line 267: like the deletion found in SARS-CoV-1 and SARS-CoV-2 that occurred very soon after their introduction in humans. Include a reference

Line 295. Reference is missing

Lines 263-294: Explain first HuCCoV_Z19Haiti and the close related  Malaysian CCoV-HuPn-2018 (Both from canines). Finally  explain PDCoV infection in humans.

Lines 301-309: This paragraph needs more references to support discussion

For this reviewer the Conclusion seems to be poor.

Author Response

Point by point response

#Reviewer 1

The review paper by Buonavoglia et al., present an updated overview of the role of Canine Coronaviruses as zoonotic emerging viruses. The low number of cases reported make the paper more a Commentary-like paper than a full review. The paper needs some graphics material  to explains different point of view of Authors.

  1. Line 28: Coronaviridae Family

R.: The test was modified

  1. Line 29. One of the largest. Specifically the GammaCoV are the largest.

R.: It was specified that GammaCoV are the largest.

  1. Line 36. 5’two-third? – One third. Use better a most accurate and scientific  description

R.: The text was modified

  1. Canine coronavirus: light and shadow: A graphic scheme is highly recommended for explain recombination events or a phylogenetic diagram indicating CCoV type IIa and CCoV type IIb and variant third subtype.

R.: As suggested, figure 1 was added to explain recombination events.

  1. Line 174-175. Clinical disease caused by CCoV is also related to the virus strain .. please correlate to the previous paragraph.

R.: The text was modified to better correlate the sentence to the previous paragraph.

  1. Line 194: Type 1 vs Type-I

R.: The text was modified

  1. 3. Pathobiology of CCoV infection in dogs: Please split in two or three paragraphs . Too much information for just one paragraph

R.: The paragraph was divided in two sub-paragraphs (3.1. Clinical signs; 3.2. Emerging CCoV strains and associated diseases)

  1. Line 199. The low detection rate in dogs cannot be due to the low stability of the enveloped virus in the environment. Perhaps low detection in the environment or low stability of the RNA in the labs.

R.: The text was modified as suggested.

  1. Please mention on this paragraph other coinfections whit enteric viruses such Canine Adenovirus, canine Circovirus, etc etc.

R.: Coinfections with other enteric viruses was reported in the text.

  1. Line 243: delete: Despite the labile and unstable nature of RNA viruses.

R.: the sentence was deleted.

  1. Line 249. Reference is missing

R.: References were added.

  1. Line 256 -257. Reference is need to support this commentary

R.: Reference was added.

  1. Line 267: like the deletion found in SARS-CoV-1 and SARS-CoV-2 that occurred very soon after their introduction in humans. Include a reference

R.: Reference was added.

  1. Line 295. Reference is missing

R.: Reference was added

  1. Lines 263-294: Explain first HuCCoV_Z19Haiti and the close related  Malaysian CCoV-HuPn-2018 (Both from canines). Finally  explain PDCoV infection in humans.

R.: The paragraph was revised as suggested.

  1. Lines 301-309: This paragraph needs more references to support discussion. For this reviewer the Conclusion seems to be poor.

R.: As suggested, the conclusion section was rephrased.

Reviewer 2 Report

In this mini-review, the authors intend to summarize the emergence of highly pathogenic CoVs in humans to underscore the significant threat that CoVs spillovers pose to humans and how we should mitigate this hazard.

Several suggestions:

1.      It is suggested to draw figures to depict the recombination events that occurred in the emerging coronavirus variants mentioned in the manuscript.

2.      It is suggested to draw the figures to depict the mutations of CCoVs (especially in the S gene) found in humans, such as HuPn-2018 strain and HuCCoV_Z19Haiti.

3.      Line 12, it is suggested to add [in dogs] after [gastroenteritis].

4.      Line 98, it is suggested to add the full names for [RT-PCR] and [RT-qPCR] when mentioned first-time.

5.      Line 236, lines 252-254, the full names for SARS-CoV 1, MERS-CoV and SARS-CoV-2 should appear in line 236.

Author Response

Reviewer#2

In this mini-review, the authors intend to summarize the emergence of highly pathogenic CoVs in humans to underscore the significant threat that CoVs spillovers pose to humans and how we should mitigate this hazard.

  1. It is suggested to draw figures to depict the recombination events that occurred in the emerging coronavirus variants mentioned in the manuscript.

R.: As suggested, a figure to depict the recombination events that occurred in emerging CCoVs was added (Figure 1)

  1. It is suggested to draw the figures to depict the mutations of CCoVs (especially in the S gene) found in humans, such as HuPn-2018 strain and HuCCoV_Z19Haiti.

R.: As suggested, a figure to depict the mutations of CCoVs, in the S gene, found in HuPn-2018 and HuCCoV_Z19Haiti was added (Figure 2).

  1. Line 12, it is suggested to add [in dogs] after [gastroenteritis].

R.: The text was modified as suggested.

  1. Line 98, it is suggested to add the full names for [RT-PCR] and [RT-qPCR] when mentioned first-time.

R.: The full name of RT-PCR and RT-qPCR were added

  1. Line 236, lines 252-254, the full names for SARS-CoV 1, MERS-CoV and SARS-CoV-2 should appear in line 236.

R.: The full name for SARS-CoV-1 was reported when first mentioned (lines 260 new text)

Reviewer 3 Report

In this study, the authors review the pathabiology of canine coronavirus (CCoV) and suggest the importance and potential threat of the highly pathogenic CoVs in humans. The manuscript is well written and provide sufficient information regarding the current knowledge of CCoV. It is suggested that if the authors can provide the figures or tables in the manuscript, it would help the reader to further understand the importance of the review and would draw more attention for broad of readers. I would suggest that the manuscript can be accepted after adding the figures. 

Author Response

Reviewer#3

In this study, the authors review the pathabiology of canine coronavirus (CCoV) and suggest the importance and potential threat of the highly pathogenic CoVs in humans. The manuscript is well written and provide sufficient information regarding the current knowledge of CCoV. It is suggested that if the authors can provide the figures or tables in the manuscript, it would help the reader to further understand the importance of the review and would draw more attention for broad of readers. I would suggest that the manuscript can be accepted after adding the figures. 

R.: As suggested, two figures were added.

Round 2

Reviewer 1 Report

The corrected version is suitable.

Reviewer 2 Report

Issues raised previously have been addressed in this revised manuscript.